# Assessment of Zoonotic Risk following Diagnosis of Canine Tularemia in a Veterinary Medical Teaching Hospital

**DOI:** 10.3390/ijerph19042011

**Published:** 2022-02-11

**Authors:** Lynelle R. Johnson, Steven E. Epstein, Jonathan D. Dear, Barbara A. Byrne

**Affiliations:** 1Department of Medicine and Epidemiology, University of California-Davis, 2108 Tupper Hall, Davis, CA 95616, USA; jddear@ucdavis.edu; 2Department of Surgical and Radiological Sciences, University of California-Davis, 2112 Tupper Hall, Davis, CA 95616, USA; seepstein@ucdavis.edu; 3Department of Pathology, Microbiology, and Immunology, University of California-Davis, 4206 VM3A, Davis, CA 95616, USA; bbyrne@ucdavis.edu

**Keywords:** zoonoses, One Health, infectious diseases, animal sentinels, epidemiology, vector-borne disease

## Abstract

Tularemia is a rare zoonotic disease found worldwide. The agent responsible for disease, *Francisella tularensis*, is one of the most highly infectious pathogens known, one that is capable of causing life-threatening illness with inhalation of <50 organisms. High infectivity explains concerns of its use in bioterrorism. This case describes a 4-year-old male neutered Australian shepherd presented for evaluation of hyporexia and fever. Physical examination revealed marked enlargement of the right superficial cervical lymph node. Tularemia lymphadenitis was diagnosed by lymph node aspiration cytology and culture. Public health officials were advised of the isolation of this zoonotic pathogen, and contact tracing was instituted. Seven individuals associated with the aspiration event were screened for tularemia and treated with prophylactic ciprofloxacin. All were negative, and none became sick. The dog was treated with doxycycline for 3 weeks, and clinical signs and physical examination abnormalities were resolved fully. The owner, a solid organ transplant recipient, was also screened for disease and received prophylactic doxycycline due to a history of shared exposure. The owner remained well throughout the course of his dog’s disease and has heightened awareness of potential zoonoses. This case highlights the importance of animals as a sentinel for human health threats and for coordination of human and veterinary care.

## 1. Introduction

*Francisella tularensis* is a Gram-negative coccobacillus responsible for the zoonotic disease known as tularemia. Tularemia is rare in the United States, with ≈250 cases reported in humans annually and a primary distribution in the south-central US and Pacific northwest [1]. In 2019, countries from the European Union reported ≈1500 human cases of tularemia, with the majority (56%) from Sweden, followed by Norway, where it is spread primarily through mosquito bites [2,3]. Ingestion of contaminated water is sometimes implicated in the spread of disease in Europe. In an outbreak of tularemia on the east coast of the USA, infected individuals were most likely exposed through mowing and brush cutting [4], although tularemia is spread by tick bites in up to 69% of human cases [5]. Tularemia is apparently less common in tropical regions and the southern hemisphere, but it has recently been reported in Australia, potentially related to exposure of a skin wound to a fish carcass or by inhalation of organisms in an air conditioning unit [6].

Tick vectors for tularemia include the dog tick (*Dermacentor variabilis*), Rocky Mountain wood tick (*D. andersoni*), and lone star tick (*Amblyomma americanum*), although transmission can also occur by fly bites, inhalation, or ingestion or handling of an infected animal carcass, particularly rodents and rabbits. In Australia, *Francisella tularensis* was isolated from ring-tailed possums that died of necrotizing enteritis or hepatitis [7], and human cases in the state of Tasmania were considered likely to have resulted from exposure to infected possums.

Clinical signs of tularemia in humans typically develop within 3 weeks of exposure (usually 3–6 days) and include fever, skin ulcers, lymphadenopathy, and pneumonia, depending on the route of exposure. Inhalational exposure resulting in pneumonia is considered the most serious and has a mortality rate in excess of 50% in the absence of treatment [8]. *Francisella tularensis* is highly infectious, with inhalation of 1–50 organisms capable of causing disease [9,10], and it has been considered an important weapon in bioterrorism [11]. In contrast, the infectious dose of anthrax, another bioterrorism agent, that can cause in infection in 50% of susceptible humans is approximately 11,000 spores [12].

Of domestic animals in the United States, tularemia is more commonly diagnosed in cats than in dogs. A small study from the northeastern US found 12–24% seropositivity in privately owned cats depending on the diagnostic test used [13], and a separate study found that ≈8% of human cases were linked to bites from infected cats [14]. This parallels data from a case–control study in Sweden where owning a cat was found to be a risk factor for human tularemia infection during a 2000 outbreak, with an odds ratio of 2.5 [3]. In contrast, *Francisella tularensis* infections in domesticated dogs are very rare, and direct transmission to humans has been documented in <2% of cases [15]. There are minimal reports on clinical manifestations of disease in dogs.

Zoonotic diseases can be spread from humans to animals, from animals to humans, or shared by both animals and humans through common exposure. Thus, zoonotic diseases are fundamental to the concept of One Health, the interdependence of human, animal, and environmental health. In the case of human tularemia, animal to animal exposure is unlikely but shared environmental exposure or contact with the organism in the laboratory or through secretions can lead to serious disease. The case description herein reveals the importance of animals as a sentinel of human disease, the potential hazards that can be encountered when performing routine procedures on animal patients, and the methodology used to evaluate risks of disease through coordination of academic and state veterinarians with physicians.

## 2. Case Description

A 4-year-old male neutered Australian Shepherd was presented for evaluation of a 1 week history of lethargy, reduced activity, and mild hyporexia. The dog had been adopted at 6 months of age and had had the right pelvic limb amputated, presumably due to a vehicular accident. At 1.5 years of age, the dog was diagnosed with sinonasal aspergillosis, which resolved after three episodes of debridement and topical infusion of clotrimazole, although marked turbinate destruction with erosion of the vomer bone occurred as a consequence of the infection. Three months prior to presentation, the dog was diagnosed with keratoconjunctivitis sicca and an indolent ulcer.

On physical examination, the dog was bright and alert with mild elevation of rectal temperature (103.4 °F, reference interval (RI) 99.5–102.5 °F; 39.7 °C, RI 37.5–39.2 °C). He had appropriate nasal airflow bilaterally and sneezed several times during the evaluation, but no coughing was noted, and no abnormalities were detected on thoracic auscultation. Changes to the left eye were consistent with previous findings including mild mucoid discharge, corneal fibrosis and melanosis with superficial corneal vessels, mild to moderate conjunctival hyperemia, and intermittent blepharospasm. His right eye was unremarkable. He had a firm, enlarged (9 × 9 cm) right superficial cervical lymph node with no other peripheral lymph node enlargement. No abdominal masses or organomegaly were palpated.

Differential diagnoses for the enlarged lymph node included reactivity due to local inflammation, infection, or neoplasia. Fine needle aspiration of the lymph node was performed by a fourth year veterinary student and clinician in an open animal treatment room within the hospital. One sample was prepared for rapid cytologic assessment in the treatment room by the student. One sample of aspirated material was placed on glass slides and air-dried, and then slides were dipped five times in a methanol-based fixative solution. The slides were then dipped five times in Diff-Quik solution I (eosinophilic) made up of xanthene dye, followed by five immersions in Diff-Quik solution II (basophilic) composed of thiazine dye, methylene blue, and azure A (Cambridge Diagnostic Products, Inc., Fort Lauderdale, FL, USA). Slides were rinsed in water, then allowed to dry prior to evaluation under oil immersion.

Three additional aspirates were submitted to the hematology laboratory for official cytologic assessment. The laboratory technician working on a benchtop under biosafety level (BSL) 1 conditions and wearing standard personal protective equipment (PPE) depressed material onto cover slips for drying, followed by Wright–Giemsa staining using an automated cell stainer (Model 7151 Wescor Aerospray Hematology Pro, ELITech Bio-Medical Systems, Logan, UT, USA).

Further assessment of the dog’s systemic illness included evaluation of a complete blood count, which showed mild neutrophilic leukocytosis (WBC 14,500 cells/µL, RI 6–13,000 cells/µL with 11,450 neutrophils/µL, RI 3–10,500 neutrophils/µL). Biochemical panel revealed a mild hyperglobulinemia (32 g/L, RI 17–31 g/L) and mildly elevated alkaline phosphatase activity (120 IU/µL, RI 14–91 IU/µL). Serum urea nitrogen was low at 3.57 nmol/L (RI 3.9–11.8 nmol/L), as was creatinine at 53.4 µmol/L (RI 70.7–97.2 µmol/L). Urinalysis was within normal limits, and urine cultures for aerobic and fungal organisms were negative. A serum *Aspergillus platelia* galactomannan enzyme immunoassay (Bio-Rad, Hercules, CA, USA) was negative. Thoracic radiographs revealed mild mineralization of the bronchial walls, normal pulmonary vasculature, and no pulmonary nodules or parenchymal abnormalities. Abdominal ultrasound showed incomplete urinary bladder distension as well as urinary bladder debris with a mixed character and mineralized sand.

Lymph node aspiration cytology demonstrated marked mixed inflammation (largely pyogranulomatous) and lymphoid reactive hyperplasia (slide not available for images due to the zoonotic nature of the organism isolated). Two additional aspiration samples were then obtained and submitted to the clinical microbiology laboratory for aerobic, anaerobic, mycobacterial, and fungal cultures. The sample was inoculated onto chocolate, 5% sheep blood, MacConkey, inhibitory mold agars (SBA, Mac, IMA; Hardy Diagnostics, Santa Maria, CA, USA) and pre-reduced anaerobe systems Brucella blood agar (PRAS Brucella; Anaerobe Systems, Morgan Hill, CA, USA) in a biosafety cabinet by a laboratory technician working under BSL 2 conditions with proper PPE. Chocolate, SBA, and Mac agars were incubated at 35 °C in a 5% CO_2_ atmosphere, whereas the PRAS Brucella agar was incubated at 35 °C under anaerobic conditions; the IMA plate was incubated at 30 °C at room atmosphere. Plates were examined daily for growth. After 72 h of incubation, scant (1+) growth was identified on chocolate agar. A second technician opened the taped chocolate agar plate to perform an extended direct transfer of the isolate to a target for matrix-assisted laser desorption/ionization–time of flight mass spectrometry (MALDI-TOF MS; Biotyper, Bruker Daltaonics, Billerca, MA, USA) in order to provide identification of the organism through analysis of the cellular proteome. This procedure was performed on an open bench under BSL 2 conditions using standard PPE.

Mycobacterial and fungal cultures were subsequently reported as negative. MALDI-TOF MS returned a result of *F. tularensis* with an identification score of 2.30, considered confident at the species level. At this point, all further work in the microbiology laboratory was halted, and the isolate was transferred to the Sacramento County Public Health Laboratory, which is part of the Laboratory Response Network for bioterrorism pathogens. Additional assessment was performed by PCR, and growth on cystine-enriched medium confirmed identity of the isolate as *F. tularensis*. Finally, direct fluorescent antibody testing was performed using a polyclonal rabbit anti-*Francisella tularensis* antibody targeting the outer membrane proteins and lipopolysaccharides of the cell wall, yielding positive results.

On the basis of the known susceptibility pattern of *Francisella tularensis*, we started the dog of this report on doxycycline 100 mg by mouth twice daily for three weeks, pending additional testing. Within two days, the owner indicated that the dog was feeling better. Follow-up evaluation three weeks later revealed that the dog was normothermic and the right prescapular lymph node was normal in size. The dog remains clinically well, as do the owner and individuals at the veterinary hospital.

Per protocol, isolation of this bioterrorism pathogen was reported to the Centers of Disease Control (CDC) in Atlanta, GA, USA, and the California Department of Food and Agriculture, and due to the zoonotic nature of tularemia, contact tracing within the hospital was performed. We reported a total of seven individuals potentially exposed to aerosolizing procedures to the Yolo County Health and Human Services Agency. Four members of the clinical staff were directly involved with the patient, lymph node aspiration, or cytologic preparation, and three were laboratory personnel. All were referred to the UC Davis Occupational Health Services for acute and convalescent (three weeks post exposure) serology to detect a rise in IgG or IgM titers. All individuals were advised to monitor for signs of illness including fever, chills, headache, and respiratory and gastrointestinal signs. The clinician and primary student who both performed lymph node aspiration and cytologic assessment were treated prophylactically with 500 mg ciprofloxacin twice daily for two weeks. Laboratory personnel were also administered prophylactic antibiotics (pending results from Occupational Health). Individuals in the radiology and ophthalmology services that had been in contact with the dog but not involved with lymph node aspiration were advised of the diagnosis of a potentially zoonotic infection in the dog and sent the CDC tularemia fact sheet, but after consultation with the California Department of Public Health, these individuals were not considered at risk for infection.

The owner of the infected dog was a solid organ transplant recipient on immunosuppressive agents and was considered of low risk of disease from direct exposure to the dog but of relatively high risk of disease because of a shared environment. The owner reported that he and the dog had been hiking in Humboldt County, a coastal region on the California–Oregon border, in early August, approximately three weeks prior to presentation. The owner did not report recovery of ticks from his dog but had found several unattached ticks on his lower leg region; therefore, he was advised to contact his physicians. His transplant and infectious disease physicians discussed the risk of zoonosis with the supervising veterinarian, performed serologic testing to assess potential exposure, and prescribed prophylactic antibiotics (doxycycline) for two weeks.

Review of the electronic medical database at the University of California Veterinary Medical Teaching Hospital identified eight additional instances between 1989 and 2021 when *F. tularensis* was isolated by the microbiology laboratory, including five primates, one other dog, one squirrel, and one mouse. One of the primates, a Capuchin monkey from a rescue organization that had consolidating pneumonia, was diagnosed ante-mortem in 1999 and was immediately euthanized. The dog presented in the current case report was the only one of these animals to have an ante-mortem diagnosis, receive appropriate treatment, and survive for follow-up.

## 3. Discussion

*Francisella tularensis* is highly infective, and while individual variability in susceptibility is recognized, it has been estimated that inhalation of <10 organisms could lead to clinical manifestations of tularemia in a large proportion of individuals [9,10,14]. Despite this, person-to-person transmission is uncommon [11], and, presumably, transmission of infection from the dog described here to its human owner would be considered equally uncommon or unlikely. However, the potential for zoonoses due to a shared environment was considered high. In a previous study of dog to human transmission [15], contact with dog saliva was the most likely means of spread, followed by contact with a dead animal, and finally by exposure to a tick from the dog, which was the major concern regarding joint exposure in this case. Most dogs in that report [15] were not clinically ill, which differs from the current report.

The low dose of *F. tularensis* needed for infectivity established the organism as an acceptable weapon for bioterrorism, and it was likely used as such in World War II [9,10]. Because of this level of infectivity and low infectious dose, laboratories isolating or performing research on *F. tularensis* are required to be BSL 3-certified. Work with BSL 3-categorized organisms must be registered with government agencies, with all interactions performed behind a set of self-closing, locking doors that separate the lab from general hallways. Laboratory personnel are sometimes immunized against microbes being evaluated. Obviously, a suspicion of infection with a serious microbe must be in place for microbiologic tests to be performed under BSL 3 conditions. Unfortunately, this was not the case here, resulting in contact tracing and post-exposure prophylactic therapy.

Slide preparation for cytologic assessment by clinicians and students has been performed in an open environment routinely in our hospital and likely has been considered standard activity in most veterinary hospitals around the world. The personnel involved in lymph node aspiration in this case were wearing protective clothing typically used in the hospital while seeing patients during the COVID-19 pandemic, including surgical face masks and lab coats or scrubs; however, gloves were not worn, and in non-COVID-19 times, face masks would not typically be worn. This has been considered standard procedure and of no particular concern for potential infectivity from dogs not suspected of infection with a zoonotic disease until the current case. Since this episode, all personnel working in the hospital have been re-educated on the potential for exposure to potential zoonosis and possible methods of exposure. While it was clear that appropriate protocols had been followed in this highly unusual case, all personnel expressed a renewed understanding of the need to use caution when handling any animal, whether ill or not.

Fortunately, this case did not expose a large number of personnel to potential infection, and the isolate was identified as a likely zoonotic agent <72 h after lymph node aspiration. The microbiologist (B.A.B.) and infectious disease control officer (S.E.E.) rapidly implemented tracing of contacts and referred all individuals to occupational or student health for appropriate screening and prophylaxis. Because the owner was known to be an individual at high risk for disease due to immunocompromise, rapid contact was made to ensure prophylaxis and also to assure the owner that the dog was not a risk for infection. Tularemia has been reported in individuals with solid organ transplants on immunosuppressant medications, and many have had a good outcome with appropriate treatment [16]. However, cases often are challenging to diagnose given lack of recognition of the rare zoonotic diseases and the need for molecular diagnostics in some cases [16]. In this case, owner risk for infection was low; however, education on the risk for zoonotic and environmental diseases in general was instituted to ensure future health.

Although the incidence of tularemia has decreased markedly since the 1950s, hundreds of human cases in the USA and across the world are documented each year. While the geographic distribution of human tularemia cases has been relatively static since reporting began, continued vigilance is required to determine the epidemiological importance of specific pathogens and to identify geographic regions of infection. With many infectious diseases, companion animals can serve as sentinels of disease and can be helpful in identifying new ecological niches for a variety of diseases [17].

In summary, tularemia in humans remains a rare condition in the United States and worldwide; however, this case illustrates the important role that animals play in highlighting the continued presence of dangerous infective organisms. Fortunately in this case, the dog had a good outcome because of the localized nature of infection, the lack of pneumonic signs, and institution of prompt medical therapy. All in-contact personnel remained healthy but had increased vigilance when performing daily tasks. The owner was deemed non-infected but now has a better understanding of the importance of zoonotic disease.

## 4. Conclusions

This paper describes the unusual case of lymphadenitis in a dog caused by *Francisella tularensis,* a disease of significant zoonotic importance due to the shared environment of man and dog. This report highlights the critical role of animals as sentinels of human disease and the need for close collaboration between animal and human doctors to ensure the health of both species, as well as to educate humans in the impact of zoonotic diseases on everyday life. This case also illustrates the potential hazards that can be encountered when performing routine diagnostic procedures on animals and emphasizes the need for diligence to avoid exposure to zoonotic agents.

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
