# Peer review of "Assessment of Zoonotic Risk following Diagnosis of Canine Tularemia in a Veterinary Medical Teaching Hospital"

_ijerph, 2022, doi:10.3390/ijerph19042011_

Round 1

Reviewer 1 Report

In this manuscript, the authors  reported a canine tularemia case. In general, the report is valuable for basic and clinical research. I only have two comments

  1. It would be appreciated if the authors includes some images or results to support the conclusions.
  2. 2.In line 178-181, please include the reference

Author Response

Reviewer :  General comment to Authors:

I have read this paper with pleasure. This is an interesting and informative case in addition to be important for raising awareness on potential zoonotic risks that can be closer than many of us realize. I would always support and applaud reporting on such. However, I would like to see a bit more discussion developed around surveillance, and linking human and veterinary medicine while confronting zoonosis, and also around zoonotic risks from companion animals. This I think is necessary to better meet the aim of the paper in terms of presenting collaborative risk evaluation from academic, human and veterinary medicine point of view.  Therefore, I would recommend it to be published after minor revisions. I have a few comments and suggestions that I think could enhance the interest of the readership.

  • Thank you for your complimentary comments and for your valuable suggestions on improvements to our paper. We agree that more information on the One Health aspects would be of value in this paper however as veterinarians, we are not suited to address human health issues any more than has been reported. We have re-emphasized the need for collaboration between human and animal doctors in practice and academia and public health throughout the manuscript, as well as in the newly constructed conclusion.  Your specific recommendations have been addressed as indicated below. 

Specific comments and suggestions:

Abstract

Lines 14-15: The highly infectious pathogen is not self-explanatory for bioterrorist concern. What really explains this concern and what would be good to add is that it is actually one of the most infectious pathogen known, and inhalation of as little as few bacteria can cause disease. Francisella tularensis has substantial capacity to cause serious illness and death.

  • Thank you for the suggestion. This has been modified to read, “The agent responsible for disease, Francisella tularensis, is one of the most highly infectious pathogens known, capable of causing life-threatening illness with inhalation of <50 organisms. High infectivity explains concerns of its use in bioterrorism.”  This is also highlighted later in the manuscript where more space is available.

Lines 26-27: I think would be beneficial to change disease into health and add threats at the end of the sentence. … for human health threats.

  • Done, thank you.
  1. Introduction

In the first paragraph you mentioned the cases in Europe and US but only for some cases you mention the means of spread. And then in the second paragraph you repeat the means for US only. Consider mentioning the ways of spreading in the first paragraph already for all mentioned countries and build the second paragraphs around important vectors and ways of exposure before getting into symptoms. In my eyes, this would make better flow.

  • Thank you, we have revised paragraphs 1 and 2 of the introduction and agree that the flow is improved.

Line 50-51: Consider being more specific. Highly infectious is vague when not explained, in comparison or illustrated by some example.

  • This has been altered to read “Francisella tularensis is highly infectious, with inhalation of 1-50 organisms capable of causing disease [9,10] and has been considered an important weapon in bioterrorism. [11] In contrast, the infectious dose of anthrax, another bioterrorism agent, that can cause in infection in 50% of susceptible humans is approximately 11,000 spores. [12]”

Lines 65-66: sentinel of human disease, not for human disease. Please correct throughout.

  • Done
  1. Case description

Lines 153, 156 and 157: consider spelling numbers less than 10

  • Done

  1. Discussion

Lines 189-190: Why you say that the transmission from the dog to human would be very unlikely? You mention risk in other places while describing the case, mention shared environment and possible exposures. The transmission is uncommon, but how you assess unlikely, and then very unlikely?

  • Thank you for pointing out our poor sentence construction. The literature states that human to human transmission is uncommon therefore we have referenced that statement and rewritten those lines as “Despite this, person to person transmission is uncommon, [11] and presumably, transmission of infection from the dog described here to its human owner would be considered equally uncommon or unlikely. However, the potential for zoonoses due to a shared environment was considered high.”

Lines 206-217: Could you develop on appropriate screening and prophylaxis with occupational risks?

  • This would be a valuable addition to the literature however as veterinarians, we are not trained to develop these tools for use in humans. We have re-emphasized the importance of diligence during potential zoonotic exposures throughout the manuscript.

If the word limit allows, I would like to encourage addressing in the discussion the introduced system of diagnostic and prophylactic tools in terms of confronting the risk, diagnosing and securing health and occupational care for risk group etc. Please see my suggestion at the beginning of my review.

  • See comments above.

  1. Conclusions

They should be number 4, not 5. But I also do to recognize them as conclusions. This is not what comes out based on your case study. What you wrote is a summary, not conclusions, and should stand as the last paragraph of discussion.

  • Thank you for this correction. A new conclusion has been written “This paper describes the unusual case of lymphadenitis in a dog caused by Francisella tularensis, a disease of significant zoonotic importance due to the shared environment of man and dog. This report highlights the critical role of animals as sentinels of human disease and the need for close collaboration between animal and human doctors to ensure the health of both species, as well as to educate humans in the impact of zoonotic diseases on every day life.  This case also illustrates the potential hazards that can be encountered when performing routine diagnostic procedures on animals and emphasizes the need for diligence to avoid exposure to zoonotic agents.”

Reviewer 2 Report

Please see my comments and suggestions in the attached file

Author Response

Reviewer:  In this manuscript, the authors reported a canine tularemia case.  In general, the report is valuable for basic and clinical research.  I only have two comments.

  • Thank you for your valuable comments in the review of our manuscript.

It would be appreciated if the authors includes some images or results to support the conclusions.

  • Unfortunately, the cytology slide from the lymph node aspirate demonstrating Francisella organisms was destroyed given the zoonotic nature of the pathogen identified and an image cannot be provided. This has been included in the text. We are uncertain what results could be illustrated but if the reviewer has recommendations, we would be happy to attempt to locate suitable images.

In line 178-181, please include the reference. 

  • The source of this information has been identified by modifying the sentence to read, “Review of the electronic medical database at the University of California Veterinary Medical Teaching Hospital identified eight additional instances between 1989 and 2021 when F. tularensis was isolated by microbiology laboratory including five primates, one other dog, one squirrel, and one mouse.”

Reviewer 3 Report

The paper “Assessment of Zoonotic Risk Following Diagnosis of Canine Tularemia in a Veterinary Medical Teaching Hospital” submitted to International Journal of Environmental Research and Public Health investigates the case of tularemia as a rare zoonotic disease on the example of a 4-year-old male neutered Australian shepherd.

It is important that the authors highlighted the often overlooked fact that zoonotic diseases can be spread from humans to animals, from animals to humans, or shared by both animals and humans through common exposure.

The article is appropriate for International Journal of Environmental Research and Public Health and is written in line with the journal’s guidelines, but minor corrections are necessary. These are as follow:

1/ The paper lacks a thorough description of a step-by-step protocol for dealing with tularemia infection. Such a protocol could be utilitarian in nature.

2/ The 'Case Description' section can be divided into case description/preventive action and future prospect.

3/ There is no discussion of other cases when dogs were infected by tularemia, either from USA or Europe (Sweden, Norway and others).

In conclusion, the improved paper should meet the stated goal of revealing the importance of animals as a sentinel for human disease, the potential hazards that can be encountered when performing routine procedures on animal patients, and the methodology used to evaluate risks of disease through coordination of academic and state veterinarians with physicians. For this reason, should be of interest to people dealing with zoonotic diseases.

Author Response

Reviewer :  The paper “Assessment of Zoonotic Risk Following Diagnosis of Canine Tularemia in a Veterinary Medical Teaching Hospital” submitted to International Journal of Environmental Research and Public Health investigates the case of tularemia as a rare zoonotic disease on the example of a 4-year-old male neutered Australian shepherd.

It is important that the authors highlighted the often overlooked fact that zoonotic diseases can be spread from humans to animals, from animals to humans, or shared by both animals and humans through common exposure.

  • Thank you for your suggestions for improvements to the manuscript.

The article is appropriate for International Journal of Environmental Research and Public Health and is written in line with the journal’s guidelines, but minor corrections are necessary. These are as follow:

1/ The paper lacks a thorough description of a step-by-step protocol for dealing with tularemia infection. Such a protocol could be utilitarian in nature.

  • We do not believe this single case report provides sufficient information to create a protocol for dealing with tularemia and do not believe as veterinarians, that it would be appropriate to devise a protocol for use in humans. We have added to the case description “Based on the known susceptibility pattern of Francisella tularensis, the dog was started on doxycycline 100 mg by mouth twice daily for three weeks pending additional testing.” (lines 153-154)

2/ The 'Case Description' section can be divided into case description/preventive action and future prospect.

  • We believe that we have followed journal instructions for a case report but will be happy to follow any further instructions from the editor.

3/ There is no discussion of other cases when dogs were infected by tularemia, either from USA or Europe (Sweden, Norway and others).

  • We have added to the first paragraph of the discussion “In a previous study of dog to human transmission [15], contact with dog saliva was the most likely means of spread followed by contact with a dead animal, and finally by exposure to a tick from the dog, which was the major concern regarding joint exposure in this case. Most dogs in that report were not clinically ill, which differs from the current report.” In the introduction, we have added “There are minimal reports on clinical manifestations of disease in dogs.” as lines 67-68.

In conclusion, the improved paper should meet the stated goal of revealing the importance of animals as a sentinel for human disease, the potential hazards that can be encountered when performing routine procedures on animal patients, and the methodology used to evaluate risks of disease through coordination of academic and state veterinarians with physicians. For this reason, should be of interest to people dealing with zoonotic diseases.

  • We believe the reviewer is requesting that the conclusion be rewritten to state the points raised above. As a rewrite was also advised by another reviewer, the conclusion has been restated as “This paper describes the unusual case of lymphadenitis in a dog caused by Francisella tularensis, a disease of significant zoonotic importance due to the shared environment of man and dog.  This report highlights the importance of animals as a sentinel of human disease and the need for close collaboration between animal and human doctors to ensure the health of both species, as well as to educate humans in the impact of zoonotic diseases on every day life.  This case also illustrates the potential hazards that can be encountered when performing routine diagnostic procedures on animals and emphasizes the need for diligence to avoid exposure to zoonotic agents.”